

# Minimal effects of ultraviolet light supplementation on egg production, egg and bone quality, and health during early lay of laying hens

Md Sohel Rana[1,2,3], Jonathon Clay[1], Prafulla Regmi[4] and Dana L.M. Campbell[2]

[1] Department of Animal Science, School of Environmental and Rural Science, University of New England, Armidale, NSW, Australia
[2] Agriculture and Food, Commonwealth Scientific and Industrial Research Organisation (CSIRO), Armidale, NSW, Australia
[3] Department of Livestock Services, Ministry of Fisheries and Livestock, Dhaka, Bangladesh
[4] Department of Poultry Science, University of Georgia, Athens, GA, United States of America

Corresponding authors
Md Sohel Rana,
ranasoheldvm06@gmail.com
Dana L.M. Campbell,
dana.campbell@csiro.au

## ABSTRACT

Chicken vision is sensitive to ultraviolet (UV) light containing the UVA spectrum, while UVB plays a key role in the endogenous production of vitamin $D_3$. However, commercially available light sources are typically deficient in the UV spectrum and thus may not adequately fulfill the lighting requirements of indoor-housed laying hens. We hypothesized that supplementary UVB light may improve egg production and egg quality, and bone health during early lay relative to UVA supplementation or standard control lighting. To investigate the effects of UV light supplementation, an experiment was conducted on 252 ISA Brown hens during 16 to 27 weeks of age. Birds were housed in eighteen pens (14 hens/pen) under three different light treatment groups each with six replications: (i) UVO: standard control lighting with LED white light, (ii) UVA: control lighting plus supplemental daylight with an avian bulb, and (iii) UVA/B: control lighting plus a supplemental full spectrum reptile bulb containing both UVA and UVB wavelengths. Hen-day egg production and egg quality, blood parameters including plasma Ca and P, and serum 25(OH)$D_3$, and hen body weight and external health scoring were measured at different age points; while bone quality was assessed at the end of the experiment at 27 weeks. Data were analyzed in JMP® 16.0 using general linear mixed models with $\alpha$ level set at 0.05. Results showed that UVA and UVA/B supplemented birds reached sexual maturity (50% production) 3 and 1 day earlier, respectively, than control birds. There was a trend for UV lights to increase hen-day egg production ($P = 0.06$). Among egg quality traits, only eggshell reflectivity and yolk index were affected by UV lights ($P = 0.02$ and 0.01, respectively); however, most of the egg quality traits changed over age (all $P < 0.01$). Post-hoc tests showed higher serum 25(OH)$D_3$ in the UVA/B group relative to control hens ($P < 0.05$); but there was no treatment effect on plasma Ca and P or on bone quality parameters (all $P > 0.05$). A significant interaction was observed between light treatment and age for the number of comb wounds ($P = 0.0004$), with the UV supplemented hens showing more comb wounds after 24 weeks. These results demonstrated that supplemental UVA/B light had minimal effects on egg production and egg quality, whereas, UVA/B exposure may increase vitamin $D_3$ synthesis during the early laying period. The optimum duration of exposure and level of intensity needs to be determined to ensure these benefits.

# INTRODUCTION

Light is an important factor in poultry production that influences bird behavior, production, and physiology (*El-Sabrout et al., 2022*; *Lewis & Morris, 1998*; *Oso et al., 2022*). As chicken vision is sensitive to ultraviolet (UV) light in the UVA (320–400 nm) spectrum (*Lewis & Morris, 2000*; *Prescott & Wathes, 1999a*), current UV deficient light sources used commercially may not fulfil hens' lighting requirements in indoor systems (*Lewis & Gous, 2009*; *Prescott, Wathes & Jarvis, 2003*). Furthermore, although chickens cannot visually perceive the UVB spectrum (280–315 nm), it stimulates the production of vitamin $D_3$ (Vit-$D_3$) which can increase intestinal absorption of calcium (Ca) and phosphorus (P); minerals that are important in both skeletal health and egg production (reviewed in *England & Ruhnke, 2020*; *Rana & Campbell, 2021*). Additional UVA/B light supply may have positive effects on indoor laying hens' physiology, behavior, egg production, and welfare but currently more research is needed to quantify its benefits (*Rana & Campbell, 2021*).

Light can pass through the eyes and skull to reach birds' pineal and pituitary glands and the hypothalamus, thus stimulating both retinal and extra-retinal photoreceptors (*Dawson et al., 2001*; *Mobarkey et al., 2010*; *Baxter et al., 2014*). A bird's growth and behavior is regulated by retinal photoreceptors while the extra-retinal photoreceptors can influence photo-sexual stimulation (*Mobarkey et al., 2010*; *Bédécarrats, 2015*). The hypothalomo-pituitary-gonadal (HPG) axis controls reproduction where gonadotropin-releasing hormones (GnRH-I and GnRH-II) stimulate production of luteinizing hormone (LH) and follicle stimulating hormone (FSH) to initiate sexual maturity (*Bédécarrats, 2015*). A prolonged nocturnal phase increases melatonin secretion and release of gonadotropin inhibitory hormone (GnIH). Longer spectral light (*e.g.*, red light) has greater energy to penetrate the skull, thus exerting greater impact on hens' reproductive physiology (*Reddy et al., 2012*; *Baxter et al., 2014*). However, the visual perception of UVA light is transmitted to the pineal gland, where it controls the circadian rhythm through melatonin, secretion of plasma growth hormones and GnRHs (*Rosiak & Zawilska, 2005*). Melatonin can also have impacts on calcium regulation and the interplay between the allocation of calcium to eggs or bones (*Taylor et al., 2013*).

The light-stimulated hormone control could result in effects of UV light on feed intake and body weight. *Liu et al. (2018)* reported that UVA light supplementation increased feed intake in layer chicks compared to control lighting. In other studies with hens in the peak or mid production phase, UVA decreased feed intake or feeding behavior (*Lewis, Perry & Morris, 2000*; *Rana et al., 2021*) and UVB had no effect on feed intake (*Gongruttananun, 2011*; *Lietzow et al., 2012*; *Schutkowski et al., 2013*). Aligning with feeding behavior, some studies with hens across peak-mid production have reported UVA (*Spindler et al., 2020*) and UVB (*Lietzow et al., 2012*) supplementation will increase body weight but not across all research (*Kühn et al., 2015*). Other physical effects of UV exposure have also been

demonstrated with UVA leading to increases in severe feather pecking and skin injuries in one commercial aviary study with hens up to 48 weeks of age (*Spindler et al., 2020*) although no effects of full spectrum bulbs (containing UVA), or UVB bulbs on plumage were found in a different study using experimental floor pens and hens at 26–37 weeks of age (*Kühn et al., 2019*). The inconsistent effects across studies and lack of research on hens specifically in the early laying phase support further research into UV light effects on physical health measures.

The sexual maturity in layer pullets coincides with a shift from structural to medullary bone formation which provides a usable source of calcium for egg production (*Whitehead & Fleming, 2000*). Given the high rate of lay of domesticated hens, the osteoporosis process begins between 16 and 31 weeks, with increasing calcium depletion as egg production continues (*Whitehead & Fleming, 2000*; *Cransberg et al., 2001*; *Alfonso-Carrillo et al., 2021*). The bone weakening can be more severe in caged systems where hens have their movement restricted, but prevalence of keel bone fractures in loose-housed hens can still be up to 98% by end of lay (*Wilkins et al., 2011*). Over this dynamic stage of physiological bone changes and increases in egg size through to peak lay, UVB light could stimulate intestinal Ca and P mineral absorption through Vit-$D_3$ synthesis, which may reduce bone mineral depletion (*De Matos, 2008*). UVB radiation acts to convert 7-dehydrocholesterol (7-DHC) to cholecalciferol (Vit-$D_3$) in hens' featherless skin of predominantly the legs (*De Matos, 2008*; *Schutkowski et al., 2013*). The subsequent transformation of Vit-$D_3$ into its metabolites (*e.g.*, $25(OH)D_3$ and $1,25(OH)_2D_3$) regulates blood Ca and P metabolism, leading to eggshell formation and bone mineralization (*Rana & Campbell, 2021*). While Vit-$D_3$ is typically supplied to hens in dietary form, further synthesis stimulation through supplemental UVB exposure could have positive effects on indoor hens. Previous studies in older laying hens showed that certain levels of daily UVB exposure increased bone mineral density, but at the end of the treatment period there were no significant impacts on serum $25(OH)D_3$, P, Ca, and $1,25(OH)_2D_3$ (*Wei et al., 2020*). Research on hens around mid-lay showed UVB radiation increased plasma concentrations of $1,25(OH)_2D_3$ but not Ca and inorganic phosphate, and $25(OH)D_3$ was only increased by radiation in the dietary Vit-$D_3$ deficient treatment birds (*Schutkowski et al., 2013*). Similarly, *Lietzow et al. (2012)* found UVB supplementation had limited additional effect over sufficient dietary Vit-$D_3$ provision and *Kühn et al. (2015)* observed no UVB impacts on plasma concentrations of $1,25(OH)_2D_3$. However, none of these studies investigated whether UVB light can affect bone health and relevant blood mineral measures during the dynamic early laying phase.

Similar to effects of UVB radiation on skeletal health, there is also some evidence for positive impacts of these wavelengths on egg production and egg quality in both older and younger laying hens (reviewed in *England & Ruhnke, 2020*; *Rana & Campbell, 2021*). However, across and within studies, the effects on hens during peak-late production are inconsistent on whether there is an impact of UVB exposure or not (*Kühn et al., 2015*; *Kühn et al., 2019*; *Schutkowski et al., 2013*; *Wei et al., 2020*). Limited daily exposure may have few effects additional to the Vit-$D_3$ supplied in the diet (*Lietzow et al., 2012*). To date UVA supplementation has not resulted in substantial effects on egg production and varying egg quality traits (*Hogsette, Wilson & Semple-Rowland, 1997*; *Sobotik, Nelson & Archer, 2020*;

*Spindler et al., 2020*). Currently, there is limited research on UVA or UVB effects on egg parameters specifically in laying hens in the beginning of the production cycle.

The objective of this study was to explore if there are any positive effects of UVB exposure during early lay on hens' egg production and egg quality, bone quality, and external health in indoor production systems. The study hypothesized that the supplementation of UVA/B light would enhance egg quality and bone quality over the UV deficient and UVA only lighting regimen. Based on previous literature, it was uncertain what direction of effect the treatment lights would have on external physical health of the hens including body weight. This study is part of a larger experiment trial that also looked at the effects of the three light treatments on hen behavioral measures (*Rana et al., 2021*, unpublished data).

## MATERIALS AND METHODS

The experimental design was approved by the Animal Ethics Committee of the University of New England (Approval no.: AEC21-034). The study was conducted in an indoor setting at the Rob Cumming Poultry Innovation Centre, University of New England (UNE), Armidale, NSW, Australia.

### Animals and housing management

Sixteen-week-old ISA Brown layer (*Gallus gallus domesticus*) pullets ($n = 252$) reared in a deep litter commercial facility with curtained-sides were procured and transferred to the experimental indoor setting with environmentally controlled rooms. All the birds were from the same flock and no experimental interventions were applied during the rearing period. Once pullets were fully feathered, the curtains during rearing were opened to facilitate natural air ventilation into the house, enabling some exposure to daylight. However, the exact degree of this daylight exposure was unknown to the researchers given birds were commercially acquired from a distant location. Daily bird care, housing management, and feeding, during rearing were as per standard commercial protocols in accordance with the Model Code of Practice for the Welfare of Animals: Domestic Poultry (*Primary Industries Standing Committee, 2002*). Upon arrival, the pullets were orally dosed (*via* gavage) with medicinal treatment for mites (Exzolt®) as per the research facility policy which was repeated in the following week along with an oral de-wormer (CCD® Levamisole, CCD® Animal Health, Tamworth, NSW, Australia). Birds were distributed equally among eighteen floor pens (14 pullets/pen) within three rooms (six pens/room). Pens were separated by wire panelling with shade cloth for hens' visual isolation and to prevent the scattering effect of the supplemental lights between pens. Each of the pens (3.2 m L × 1.75 m W) contained a single three-rung perch (1.07 m L × 64 cm W × 80 cm H), one rollaway nest box (34 cm L × 29 cm W × 24 cm H), one round feeder (40 cm H × 43.5 cm D × 1.36 m C), and two nipple drinkers. Resources either met or exceeded those set in the Model Code of Practice (*Primary Industries Standing Committee, 2002*) but no additional enrichments were provided in the experimental pens. Feed and water were provided *ad libitum* where a standard commercial grower crumble (Barastoc® pullet grower crumble, Melbourne, Australia) was fed on the first week of arrival, followed by commercial layer mash (Barastoc® Top Layer mash, Melbourne, Australia) for the

remainder of the study period. Wood shavings were used as floor litter (5 cm depth). All birds were visually checked at least once daily to ensure any welfare issues were immediately detected and addressed.

Pens were artificially illuminated with poultry-specific LED white light bulbs (IP65 dimmable LED bulb, B-E27-10W-5K, Eco-Industrial Supplies, China) with a light intensity of 30 lux at birds' eye height across the pens (Testboy TV 335 Digital LED Luxmeter, GmbH, Germany) when the birds were not close to or directly under the supplemental lights. The approximate hen stocking density was 2.5 hens/m². The hens were maintained on an ISA Brown specific lighting schedule (14L:10D at 16 weeks which was increased by 15 min each week until birds reached 24 weeks when the schedule was fixed at16L:8D for the remaining study period; lights on at 0500 h and off at 2100 h). The indoor temperature and humidity were mechanically controlled through an automatic ventilation system. Hens were leg-banded for unique identification to facilitate experimental measures. At 27 weeks, a subset of hens (four hens/pen; 24 hens/treatment; total 72) were sacrificed by cervical dislocation following an electrical stunning to assess the effects of light treatments on bone health. The euthanasia protocol was as per standards set by the Animal Ethics Committee of the University of New England. All remaining birds were rehomed at the end of the experiment.

## Experimental design

On the day after bird arrival, pens were divided into three different lighting treatment groups (six pens/treatment): (i) UVO (420–724 nm): control LED-white light (IP65 dimmable LED bulb, B-E27-10W-5K, Eco-Industrial Supplies, China) with no UV spectral light supplementation; (ii) UVA (315–712 nm): LED-white light with supplementation of full-spectrum avian daylight (PureSun Compact Bird Lamp, E27-20W, Arcadia, Germany) containing UVA and UVB where 3 mm glass was used directly under the bulb to filter out the UVB spectrum; and (iii) UVA/B (288–714 nm): LED-white light with supplementation of full-spectrum Exo-Terra® (Rolf C. Hagen, Montreal, Canada) pet reptile bulbs (Reptile UVB200, 25W, PT2341) containing both UVA and UVB spectrums (see Figs. S1–S3 for spectrographs of the control and treatment bulbs). Location of treatment pens within each room were balanced across the three rooms. To account for increased lux from the supplemental bulbs in the treatment pens, the light intensity of the control pens was increased by adding an LED filament warm-white light (Edison Screw LED GLS Filament Globe, E27-4W, Mirabella, VIC, Australia) (Fig. S4). All supplemental bulbs were within Arcadia ceramic reflector clamp lamps E27 (200 mm, RARM160X) suspended 76 cm above the floor on one side of the pens. Thus, birds could choose to be located under the standard lighting or the illuminated area under the treatment light. Standing chicken eye height was approximately 40 cm (5 cm wood shavings) away from the light source but the chickens were closer if they stretched their necks (Fig. 1).

To facilitate UV irradiance exposure to the birds' legs and feet (the greatest areas of UV radiation absorbance; *Schutkowski et al., 2013*), the perch was located on the treatment light side. If the hens were mounted on the perch, their legs were approximately 30–50 cm distance from the bulb depending on their movement on the different perch rungs.

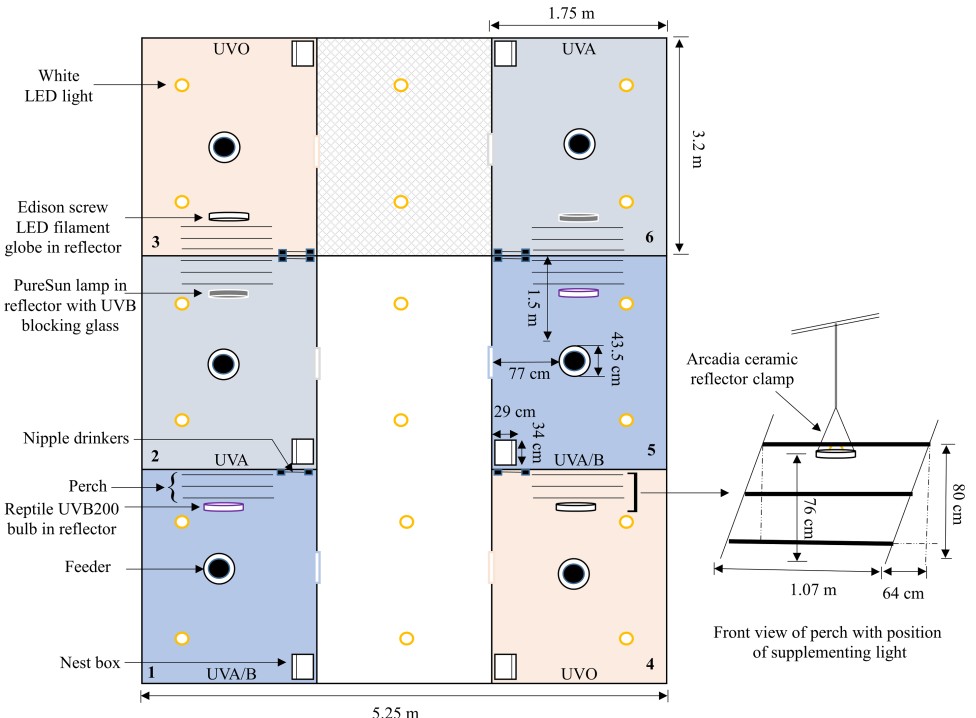

**Figure 1  A schematic of one of the three experimental rooms.** A diagram showing the placement and layout of the control (UV0) and treatment (UVA, UVA/B) pens within the room (pen positions were balanced across the three rooms). The position of the control and treatment supplemental lights along with a three-rung perch, feeder, nipple drinkers, and a nest box are indicated. Dimensions are provided but the schematic is not drawn precisely to scale.

The intensity of light and UVB irradiance was greatest directly underneath the treatment light source which then gradually decreased due to scattering as distance from the bulb increased (Table 1). A timer was set in each pen to turn the treatment lights on/off in approximate accordance with the standard lighting schedule (best match possible based on the accuracy of the analogue timers). The light intensities were measured using a lux meter (Testboy TV 335 Digital LED Luxmeter, GmbH, Gerlingen, Germany), and the UV index was measured for the UVA/B treatment using a reptile UV index meter (Solarmeter Model 6.5 R UVI Reptile, Solarmeter Australia, Noosaville, QLD) at both floor level and hens' eye height. The light spectrums and radiation irradiance were measured over an average of 10 readings using an Ocean Insight Flame-S-XR1 Spectroradiometer (200–1,025 nm, Quark Photonics, Melbourne, Australia) set with an integration time of 180,000 ms and integration range from 280 to 1,000 nm (Figs. S1–S4). The UV-filtering glass was cleaned with an alcohol wipe every 14 days to remove the dust.

## Blood sampling, plasma and serum separation

At 20, 23, and 26 weeks, blood samples were taken in the morning between 0900 and 1200 h when the majority of the birds had laid their eggs. The same randomly selected subset of hens (five hens/pen; 30/treatment) was sampled each time (identified by numbered/colored

**Table 1 Light intensity and UV index directly underneath the treatment lights in the pens.**

| Light treatment[a] | Level of light measurement | Light intensity (lux) | UV index[b] |
|---|---|---|---|
| UVO | Floor level | 77.58 ± 3.33 | – |
| | Eye height (30 cm) | 207.82 ± 9.80 | – |
| UVA | Floor level | 69.50 ± 1.77 | – |
| | Eye height (30 cm) | 174.77 ± 6.04 | – |
| UVA/B | Floor level | 75.25 ± 2.94 | 0.40 |
| | Eye height (30 cm) | 181.40 ± 10.34 | 0.90 |

**Notes.**
[a]Treatments: UVO: Control light with no ultraviolet (UV) spectral light, UVA: Control light plus UVA spectral light, UVA/B: Control light plus both UVA and UVB spectral light.
[b]The UV index was only available for the UVA/B as this can only be calculated where both UVA and B spectrums are present.

leg bands). Three-experienced operators drained approximately 2–3 ml of blood from the wing veins of each hen using a 19-gauge needle. Approximately 1−1.5 ml of blood per hen was collected into a lithium-heparin plasma separation gel vacutainer (BD Vacutainer® PST Lithium Heparin Tubes, BD 368056, Franklin Lakes, NJ, USA) to separate plasma; while the rest of the blood was collected into another vacutainer (BD Vacutainer® Serum Tubes, BD367812, Franklin Lakes, NJ, USA) to separate serum. The vacutainers were immediately stored in an ice bath and shifted to the laboratory within 3 h after collection. On the same day, the plasma tubes were spun down in a bench-top centrifuge machine (Dynamica, Velocity 30R, Livingston, UK) for 15 min at 3,000 RPM. Then the supernatant portion of plasma was poured off into 2 ml microcentrifuge tubes using a plastic pipette and stored at −20 °C until further analysis. The serum tubes were kept at room temperature for 2–3 h to let the blood cells settle and then spun down in a bench-top centrifuge machine (Elmi® Ltd, CM-6MT, Riga, Latvia) for 15 min at 3,000 RPM. The supernatant portion of serum was then transferred into 2 ml microcentrifuge tubes using a plastic pipette and stored at −20 °C until further analysis.

## Bone sampling

At 27 weeks, a total of 72 hens from the same subset of birds used for blood sampling (four hens/pen; 24 hens/treatment) were euthanized by cervical dislocation following an electrical stunning. The left tibia bone was excised and stored with flesh at −20 °C until further processing. Later, the bones were placed at room temperature overnight for thawing and then defleshed by hand using a scalpel before taking the final measures. Bones were wrapped in gauze-soaked saline to prevent drying until further procedures were conducted (see section **Bone Health Assessments**).

## Egg production and egg quality measurements

Eggs from all pens were collected daily across the study period to measure hen-day egg production (%). Egg quality was measured twice at 21 and 25 weeks for all treatment groups. All eggs of a single day were collected in the morning, transferred to an egg quality testing laboratory, and measured on the same day (except eggshell weight and eggshell thickness) by a single experimenter. The eggs were not washed but were freed from litter debris before

taking the measures. The order of quality testing within the treatment groups was not balanced across the day but all measures were taken within 4 h after egg collection. Any eggs that were significantly deformed were not included in the measurements. Egg length (mm) and width (mm) were measured using a Digital Vernier Caliper scale (Kincrome, Sydney, Australia) for calculating egg shape index (egg length divided by egg width); egg length was measured as the dimension between the poles, while the width was measured at the equator. Eggshell reflectivity (%) was then assessed using a shell reflectivity meter (Technical Services and Supplies, Dunnington, York, United Kingdom) with measurements taken in three places (equator, base and apex) and the average value used for the analyses.

Egg weight (g), eggshell breaking strength (Kgf), and other internal egg quality traits including albumen height (mm), yolk color, yolk height (mm), yolk diameter (mm), yolk index and Haugh unit were determined by using an egg multi-tester instrument (Nabel DET-6500, Kyoto, Japan). Afterward, eggshells were washed thoroughly in tap water to remove attached albumen from the shell surface and placed at room temperature for at least 48 h for air-drying before taking the eggshell weight (g) and eggshell thickness (mm). Eggshell weight (g) was measured on shells that had been washed and dried thoroughly, using an Adventurer™ Precision analytical balance (Model AX423, Ohaus®, NJ, United States). Eggshell (%) was measured as the percentage of total egg weight. Eggshell thickness (mm) was taken in three places (equator, base and apex) using a custom-built gauge, based on a Mitutoyo Dial Comparator Gauge (Model 2109A–10, Kawasaki, Japan) and the average value was used for the analyses.

## Physiological blood measurements
### Plasma Ca and P

Analysis of plasma Ca and P was performed using a Thermo Indiko™ Plus auto-analyser (Thermo Fisher Scientific, Waltham, MA, USA). Calcium concentration was determined by a dye-binding method with Arsenazo III as described by *Janssen & Helbing (1991)*; while P concentration was determined by the formation of the phosphomolybdate complex as per the method described in *Daly & Ertingshausen (1972)*.

### Serum 25(OH)D$_3$

Quantification of 25(OH)D$_3$ was performed using high-pressure liquid chromatography-diode array detector (HPLC-DAD) based on the method described in *Galunska et al. (2014)* with slight modification. A HPLC system (Dionex UltiMate 3000, Thermo Scientific, Waltham, MA, USA) equipped with UV diode array detector was used for chromatographic analysis. The separation of analytes was performed using a Thermo Scientific™ Acclaim Polar Advantage II column (C18, 5 $\mu$m, 4.6 × 250 mm). The mobile phase consisted of 70% acetonitrile, 25% methanol and 5% water and used an isocratic elution with a flow rate of 1.0 ml/min for a 20 min run time. Samples were injected at a volume of 100 $\mu$l, and detection was carried out at 265 nm. Sample preparation was adapted from *Olkowski, Aranda-Osorio & McKinnon (2003)*. In brief, 200 $\mu$l of serum samples were added to a 2 ml polypropylene tube followed by 200 $\mu$l acetonitrile. Samples were vortexed for 15 s before being centrifuged at 10,000 × g (Beckman Microfuge 16 MicroCentrifuge, Beckman Coulter, California, USA) for 20 min. Aliquots of the supernatant were pipetted into sample

vials before analysis. Standards were made from pure 25(OH)D$_3$ (Sigma-Aldrich, Saint Louis, MO, USA) dissolved in methanol and stored at −20 °C before being diluted into 0.5% BSA solution prior to extraction as per serum samples. All solvents used were HPLC grade (ChemSupply, Gillman, Australia).

## Bone health assessments
### Tibia breaking strength

The left tibia bone was subjected to testing for breaking strength (N) with an Instron® electromechanical universal testing machine (Instron® Mechanical Testing Systems, 825 University Ave, Norwood, MA, USA) set up at 300 KN load cell and 50 mm 3-point flexure test at 0.2 mm/second test speed with 20 data points per second. The Bluehill® universal software was used to record the data. The mechanical force was applied to the midpoint of the bone with a 2 cm distance between the two 50 mm height fixed points supporting the bone. All the bones were tested on same day, following the day of defleshing.

### Tibia ash percentage

Following bone-breaking strength, the remnant of the broken left tibia bone was weighed with a crucible (wet weight) and put in a forced-air oven at 105 °C (Watson Victor Ltd, Sydney, Australia) for drying for at least 16 h. Then the samples were taken from the oven and placed in a desiccator for 30–40 min. Dry matter (%) was determined by re-weighing the samples with a crucible, divided by wet weight and multiplied by 100. Then the crucibles containing the bone samples were placed in a muffle furnace (Carbolite, Sheffield, England) overnight for ashing set at 350 °C holding for 1 h, which was then increased to 600 °C holding for 2.5 h. On the following day, the crucibles containing the bone ash samples were taken from the furnace and placed in a desiccator for 30–40 min. The ash content (g) was weighed and then divided by dry bone weight and multiplied by 100 to determine the ash (%).

### Tibia mineral content

For the determination of bone mineral contents, predominantly Ca (%) and P (%) in the tibia bone, 500 mg of ash were homogenized and digested in the Milestone Ultrawave® microwave (Milestone Srl, Sorisole, Italy) with nitric acid (HNO$_3$) and then the concentration of minerals was measured by inductively coupled plasma emission spectrometer (ICP-OES) (Agilent, Santa Clara, CA, USA). In brief, 500 mg of ash sample were taken into a Teflon tube and added with 1 ml of deionised water and 4 ml of 70% HNO$_3$ solution for pre-digestion overnight. Then the Teflon tubes were loaded in a digestion rack and placed in the microwave cavity following the Ultrawave instructions for digestion time and pressure. After completion of the digestion process, the digested samples were made to approximately 25 ml total volume with deionised water for dilution giving a 16% HNO$_3$ concentration. Each sample was run in triplicate and the average was taken across the three runs to determine mineral contents. Standard and blank samples were also run for quality assurance.

## Body weight and external health scoring

At 16, 20, 24, and 27 weeks, hens underwent body weight measurements using poultry-specific electronic hanging scales (BAT1; VEIT Electronics, Moravany, Czech Republic) and external welfare scoring based on the criteria in *Tauson et al. (2005)*. This scoring system included feather loss across body parts (neck, chest, back, wing, vent, tail) and footpad lesions where a score of 1 indicated no damage, and a score of 4 the worst damage (*i.e.,* bare skin, or bumble-foot for plumage and feet, respectively). All fresh or healing comb wounds were counted with toenail length measured in mm using a seamstress tape measure. Beaks were scored as 0, 1, or 2 indicating no, mild, or moderate/severe damage, respectively and keels were palpated and scored as 1, 2, or 3 indicating no, mild, or moderate/severe damage, respectively. Any other signs of injury or illness were noted such as swollen abdomen, eye infection, respiratory issues, and toe damage. The same trained individual did all the scoring across the study.

## Data and statistical analyses

All data were analysed using JMP® 16.0 (SAS Institute, Cary, NC, United States) with $\alpha$ set at 0.05 and a trend was considered as $P = 0.05 \leq 0.10$. Data were transformed where necessary to approach normality with studentized model residuals plotted for visual inspection of homoscedasticity. Eggs were collected on a daily basis when hens commenced laying at 17 weeks of age where the timing of sexual maturity was estimated for each treatment by determining the age when hen-day egg production (%) reached 50%. The hen-day egg production was determined on a weekly basis (the total number of eggs of a respective week per pen was pooled, then divided by the total number of hens per pen $\times$ 7 days $\times$ 100). The proportional data of hen-day egg production were logit transformed after subtracting a constant of 0.03 from all data to allow transformation of proportional values '1 or >1' (100% or >100% production, $n = 8$ and 11 of 162, respectively) to approach normality. General linear mixed models (GLMMs) were applied to hen-day egg production with treatment and age (week) and their interaction as fixed effects, and room and pen as random effects. Both external and internal egg quality traits of the sampled eggs were measured twice at 21 ($n = 222$ eggs) and 25 ($n = 237$ eggs) weeks. Eggshell (%) and eggshell reflectivity (%) data were converted to proportions and logit transformed. Separate GLMMs were fitted to each egg quality variable (egg shape index, eggshell reflectivity (%), egg weight (g), eggshell weight (g), eggshell (%), eggshell breaking strength (Kgf), eggshell thickness ($\mu$m), albumen height (mm), yolk index, yolk color, Haugh unit) with treatment and age as fixed effects including their interaction; and pen and room included as random effects. Post hoc Student's t-tests were applied to the least squares means where significant or trend differences were present, with the raw values presented in the tables and graphs.

To assess the effects of light treatments on blood physiological parameters, plasma Ca and P ($n = 270$: 5 hens $\times$ 18 pens $\times$ 3 age points), and serum 25(OH)D$_3$ ($n = 191$: 5 hens $\times$ 18 pens $\times$ 3 age points where 37, 29, and 13 samples were missing due to sample integrity at 20, 23 and 26 weeks, respectively) data were analysed by fitting separate GLMMs where treatment, age, and their interaction were fixed effects, and room, pen and hen ID were random effects. Both plasma Ca and P data did not require any transformation as the data

were normally distributed while serum 25(OH)D$_3$ data were log transformed to approach normality. For bone quality analyses, the left tibia bone ($n = 72$: 4 hens × 18 pens) index was calculated using the following formula: Tibia index = (Tibia weight (g)/Body weight (g)) × 100. The percentage data of tibia ash, Ca, and P were converted to proportions and logit transformed. Tibia breaking strength and tibia ash content (g) did not require any transformation as the data were normally distributed. Data were analysed using separate GLMMs where treatment was a fixed effect, and room and pen were random effects. *Post hoc* Student's *t*-tests were applied to the least squares means where significant or trend differences were present, with the raw values presented in the tables and graphs.

Growth and the health scoring data including the body weight, beak score, number of comb wounds, keel bone score, feather scores (neck, chest, back, wing, vent, tail), and toenail length at different age points (16, 20, 24, and 27 weeks) throughout the study period for individual hens from different light treatments were compiled ($n = 1,008$ data points/welfare parameter: 252 hens × 4 age points). The comb wound count data were square-root-transformed and the toenail length data were log transformed to improve normality. For the assessment of body weight, number of comb wounds, and toenail length, GLMMs were fitted with treatment and age and their interaction as fixed effects, and room, pen and hen ID as random effects. *Post hoc* Student's t-tests were applied to the least squares means where significant differences were present, but the raw values are presented in the tables and graphs. An ordinal logistic regression was performed to analyse the beak scores and keel bone scores using treatment and age along with their interaction as fixed effects. There was no observed variation detected in the hens' feather scores and footpad lesions irrespective of the light treatments and age during the study period (all hens had full plumage coverage and no footpad dermatitis) so these data were not analyzed.

## RESULTS

### Age at sexual maturity

This study showed that birds were sexually matured (50% hen-day egg production) at 19 weeks irrespective of the lighting treatments. However, hens under the UVA and UVA/B light treatments came into sexual maturity before the control birds (3 and 1 day earlier, respectively).

### Egg production

There was a trend of increased hen-day egg production for the effect of UV lights relative to the control light (mean ± SEM: UVO: 74.15 ± 2.50%, UVA: 79.21 ± 2.50%, and UVA/B: 79.01 ± 2.50%; $F_{(2,13.04)} = 3.55$, $P = 0.06$), with post-hoc tests showing hen-day egg production under both UVA and UVA/B light treatments was higher than the control hens across the study period ($P < 0.05$; Fig. 2). As expected, the hen-day egg production significantly increased over age ($F_{(8,118.20)} = 110.95$, $P < 0.0001$; Fig. 2). There was no significant interaction between light treatment and age for hen-day egg production ($F_{(16,118.10)} = 0.53$, $P = 0.93$).

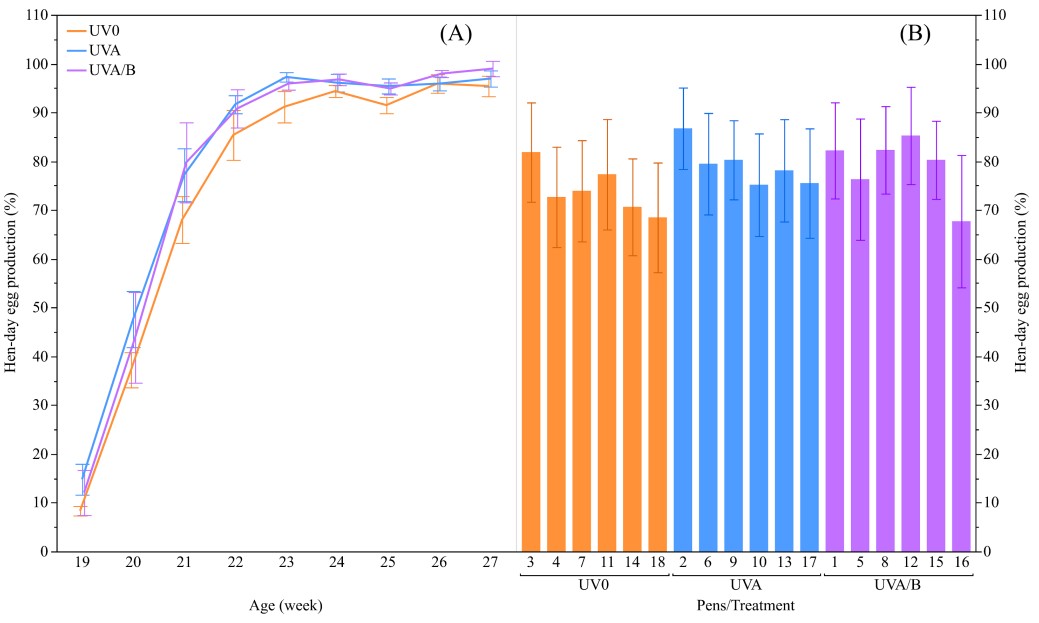

**Figure 2** **The mean ± SEM effects of light treatment (ultraviolet (UV) 0, UVA, UVA/B) on hen-day egg production (%).** (A) Between the treatment groups at different ages, (B) Across the pens within the same treatment group over the study period. Raw data are presented with analyses conducted on transformed data.

## Egg quality

The effects of light treatment, age and their interactions on egg quality parameters are shown in Table 2. There were significant main effects of both treatment lights and age on eggshell reflectivity (treatment: $P = 0.02$; age: $P < 0.0001$) and egg yolk index (treatment: $P = 0.01$; age: $P = 0.001$). The eggshell reflectance increased for the eggs under both UVA and UVA/B light treatments and the egg yolk index decreased under UVA/B light, while both of the parameters were also affected by age with increasing reflectivity and decreasing yolk index across age. Parameters including egg weight, eggshell weight, eggshell percentage, eggshell breaking strength, eggshell thickness, and egg yolk color were not affected by light treatments (all $P > 0.05$) but these traits significantly increased across age (all $P < 0.01$, Table 2). Albumen height and Haugh unit significantly decreased across age ($P < 0.0001$) but were not affected by light treatment (both $P > 0.05$). There was only a trend for an interaction effect of light treatment and age on eggshell weight ($P = 0.05$, Table 2).

## Bone quality

There was no significant effect of UV light treatments on the tibia bone index, tibia breaking strength, ash content, and ash percentage (all $P > 0.05$; Table 3). UV light also did not affect the tibia ash mineral concentrations of Ca and P (both $P > 0.05$; Table 3).

## Physiological blood measurements

The UV light treatments did not have significant effects on the plasma Ca and P concentrations measured at different age points during the early laying phase (both

 

**Table 2  Effects of UV light treatments on egg quality traits at 21 and 25 weeks.**

| Variable | Treatment[c] | | | Age | | Test statistics (df, F-ratio, P-value) | | |
|---|---|---|---|---|---|---|---|---|
| | UVO | UVA | UVAB | 21 weeks | 25 weeks | T | A | T × A |
| Egg shape index | 79.21 ± 0.17 | 79.26 ± 0.15 | 78.74 ± 0.15 | 79.12 ± 0.14 | 79.02 ± 0.14 | $F_{(2,452)} = 2.25$, $P = 0.11$ | $F_{(1,452)} = 0.15$, $P = 0.70$ | $F_{(2,452)} = 0.65$, $P = 0.52$ |
| Eggshell reflectivity (%) | 26.31 ± 0.47[b] | 27.73 ± 0.46[a] | 28.42 ± 0.46[a] | 26.37 ± 0.3[b] | 28.60 ± 0.31[a] | $F_{(2,13.05)} = 5.82$, $P = 0.02$ | $F_{(1,439.40)} = 46.57$, $P < 0.0001$ | $F_{(2,439.60)} = 1.98$, $P = 0.14$ |
| Egg weight (g) | 53.27 ± 0.51 | 53.93 ± 0.50 | 53.29 ± 0.50 | 50.98 ± 0.40[b] | 56.01 ± 0.39[a] | $F_{(2,13.65)} = 0.62$, $P = 0.56$ | $F_{(1,440.70)} = 124.24$, $P < 0.0001$ | $F_{(2,440.90)} = 0.81$, $P = 0.45$ |
| Eggshell weight (g) | 5.76 ± 0.05 | 5.75 ± 0.05 | 5.76 ± 0.05 | 5.32 ± 0.02[b] | 6.20 ± 0.02[a] | $F_{(2,453)} = 0.01$, $P = 0.99$ | $F_{(1,453)} = 338.01$, $P < 0.0001$ | $F_{(2,453)} = 3.04$, $P = 0.05$ |
| Eggshell (%) | 10.81 ± 0.11 | 10.69 ± 0.11 | 10.80 ± 0.11 | 10.44 ± 0.07[b] | 11.09 ± 0.07[a] | $F_{(2,13.28)} = 0.49$, $P = 0.63$ | $F_{(1,439.30)} = 103.22$, $P < 0.0001$ | $F_{(2,439.30)} = 1.58$, $P = 0.21$ |
| Eggshell breaking strength (Kgf) | 5.13 ± 0.10 | 5.08 ± 0.10 | 5.13 ± 0.10 | 5.01 ± 0.07[b] | 5.22 ± 0.06[a] | $F_{(2,13.25)} = 0.09$, $P = 0.92$ | $F_{(1,439.50)} = 9.46$, $P = 0.002$ | $F_{(2,439.40)} = 0.37$, $P = 0.69$ |
| Eggshell thickness (μm) | 433.61 ± 0.004 | 427.99 ± 0.004 | 434.46 ± 0.004 | 424.88 ± 0.002[b] | 439.16 ± 0.002[a] | $F_{(2,13.25)} = 0.84$, $P = 0.45$ | $F_{(1,439.30)} = 43.31$, $P < 0.0001$ | $F_{(2,439.20)} = 1.01$, $P = 0.34$ |
| Albumen height (mm) | 9.91 ± 0.14 | 9.75 ± 0.14 | 9.72 ± 0.14 | 10.06 ± 0.12[a] | 9.53 ± 0.12[b] | $F_{(2,12.54)} = 0.87$, $P = 0.44$ | $F_{(1,439.80)} = 19.73$, $P < 0.0001$ | $F_{(2,440.50)} = 1.24$, $P = 0.29$ |
| Yolk index | 0.54 ± 0.004[a] | 0.53 ± 0.004[a,b] | 0.52 ± 0.004[b] | 0.54 ± 0.003[a] | 0.52 ± 0.003[b] | $F_{(2,13.08)} = 6.01$, $P = 0.01$ | $F_{(1,441)} = 12.13$, $P = 0.001$ | $F_{(2,440.20)} = 2.04$, $P = 0.13$ |
| Yolk color | 10.08 ± 0.23 | 9.70 ± 0.23 | 9.84 ± 0.23 | 9.30 ± 0.20[b] | 10.45 ± 0.20[a] | $F_{(2,13)} = 1.35$, $P = 0.29$ | $F_{(1,440)} = 42.99$, $P < 0.0001$ | $F_{(2,440.70)} = 0.44$, $P = 0.65$ |
| Haugh unit | 100.22 ± 0.59 | 99.16 ± 0.58 | 99.41 ± 0.58 | 101.37 ± 0.47[a] | 97.82 ± 0.46[b] | $F_{(2,12.7)} = 1.11$, $P = 0.36$ | $F_{(1,439.70)} = 48.40$, $P < 0.0001$ | $F_{(2,440.10)} = 2.02$, $P = 0.13$ |

**Notes.**

[1]Treatments: UVO, Control light with no ultraviolet (UV) spectral light; UVA, Control light plus UVA spectral light; UVA/B, Control light plus both UVA and UVB spectral light; T, Treatment; A, Age. The means ± SEM are presented for each variable. Raw data are presented with analyses conducted on transformed data. Significant $P$-values ($< 0.05$) are indicated in bold.

[a,b]Dissimilar superscript letters indicate significant post-hoc differences between the groups ($P < 0.05$).

**Table 3  Effects of UV light treatments on tibia bone quality traits at 27 weeks.**

| Tibia parameters | Treatment [1] | | | Test statistics (df, F-ratio, P-value) |
|---|---|---|---|---|
| | UVO | UVA | UVA/B | |
| Tibia index | 0.68 ± 0.01 | 0.69 ± 0.01 | 0.67 ± 0.01 | $F_{(2,13)} = 1.63$, $P = 0.23$ |
| Tibia breaking strength (N) | 208.78 ± 11.70 | 214.17 ± 11.70 | 209.77 ± 11.70 | $F_{(2,13)} = 0.14$, $P = 0.87$ |
| Ash content (g) | 3.60 ± 0.07 | 3.63 ± 0.07 | 3.48 ± 0.07 | $F_{(2,13)} = 1.43$, $P = 0.28$ |
| Ash (%) | 44.07 ± 0.87 | 43.29 ± 0.87 | 43.30 ± 0.87 | $F_{(2,13)} = 0.43$, $P = 0.66$ |
| Ca (%) | 38.88 ± 1.95 | 36.92 ± 1.95 | 37.10 ± 1.95 | $F_{(2,13)} = 0.67$, $P = 0.53$ |
| P (%) | 17.11 ± 0.80 | 16.23 ± 0.80 | 16.42 ± 0.80 | $F_{(2,13)} = 0.58$, $P = 0.57$ |

**Notes.**

[1]Treatments: UVO, Control light with no ultraviolet (UV) spectral light; UVA, Control light plus UVA spectral light; UVA/B, Control light plus both UVA and UVB spectral light. The means ± SEM are presented for each variable. Raw data are presented with analyses conducted on transformed data.

**Table 4    Effects of UV light treatments on plasma Ca and P, and serum 25(OH)D$_3$ levels at 20, 23 and 26 weeks.**

| Parameters | Fixed effects | Plasma Ca (nmol/ml) | Plasma P (nmol/ml) | Serum 25(OH)D$_3$ (ng/ml) |
|---|---|---|---|---|
| **Treatment**[1] | UVO | $5.79 \pm 0.21$ | $1.72 \pm 0.05$ | $19.44 \pm 1.68^{b}$ |
| | UVA | $5.77 \pm 0.21$ | $1.72 \pm 0.05$ | $24.26 \pm 1.50^{a,b}$ |
| | UVA/B | $6.16 \pm 0.21$ | $1.86 \pm 0.05$ | $24.74 \pm 1.85^{a}$ |
| | **Test statistics (df, F-ratio, P-value)** | $F_{(2,13)} = 1.46, P = 0.27$ | $F_{(2,13)} = 2.00, P = 0.18$ | $F_{(2,11.86)} = 3.03, P = 0.09$ |
| **Age** | 20 weeks | $6.15 \pm 0.19^{a}$ | $1.95 \pm 0.04^{a}$ | $23.16 \pm 1.58$ |
| | 23 weeks | $5.65 \pm 0.19^{b}$ | $1.65 \pm 0.04^{b}$ | $22.38 \pm 1.54$ |
| | 26 weeks | $5.92 \pm 0.19^{a,b}$ | $1.70 \pm 0.04^{b}$ | $22.90 \pm 1.24$ |
| | **Test statistics (df, F-ratio, P-value)** | $F_{(2,174)} = 3.24, \boldsymbol{P = 0.04}$ | $F_{(2,174)} = 12.92, \boldsymbol{P < 0.0001}$ | $F_{(2,144.40)} = 0.09, P = 0.91$ |
| T × A | **Test statistics (df, F-ratio, P-value)** | $F_{(4,174)} = 0.97, P = 0.43$ | $F_{(4,174)} = 0.96, P = 0.43$ | $F_{(4,132.40)} = 1.05, P = 0.38$ |

Notes.

[1]Treatments: UVO, Control light with no ultraviolet (UV) spectral light; UVA, Control light plus UVA spectral light; UVA/B, Control light plus both UVA and UVB spectral light; T, Treatment; A, Age. The means ± SEM are presented for each variable. Raw data are presented with analyses conducted on transformed data. Significant $P$ values (<0.05) are indicated in bold.

[a,b]Dissimilar superscript letters indicate significant post-hoc differences between the groups ($P < 0.05$).

$P > 0.05$; Table 4). However, there was an inconsistent increase in plasma Ca concentrations over age where the Ca concentrations were significantly higher in both the initial and final measurements at 20 and 26 weeks respectively compared to the second measure at 23 weeks ($P = 0.04$; Table 4). The plasma P concentrations were significantly lower in both the second and final measurements at 23 and 26 weeks respectively than in the initial measure at 20 weeks ($P < 0.0001$; Table 4). On the other hand, there was a trend for an effect of UV lights to increase serum 25(OH)D$_3$ over control lights ($P = 0.09$; Table 4), with *post-hoc* tests showing the UVA/B group had higher 25(OH)D$_3$ concentrations than the control group ($P < 0.05$; Fig. 3). There was no effect of age on serum 25(OH)D$_3$ concentrations ($P = 0.91$; Table 4) and no interaction effects for light treatment and age on plasma Ca and P, and/or serum 25(OH)D$_3$ concentrations (all $P > 0.05$; Table 4).

## Body weight and health welfare scoring

This study did not find a significant effect of light treatment on hens' body weight across the study period ($P = 0.19$; Table 5). However, as the study period was during the hens' grower stage and early lay, body weight significantly increased over age ($P < 0.0001$; Table 5). There was no significant interaction between light treatment and age on hens' body weight ($P = 0.68$; Table 5). There was a significant interaction effect between light treatment and age on the number of comb wounds ($F_{(6,748)} = 4.15, P = 0.0004$; Fig. 4) with both UV light treatment hens showing a greater increase in comb wounds than control hens at 24 and 27 weeks. Toenail length showed a trend to decrease under control and UVA/B light treatments but significantly reduced over age and no interaction effect was observed ($P = 0.09$, $P < 0.0001$, and $P = 0.27$, respectively; Table 5). Light treatment and age affected beak score ($P = 0.09$ and $P = 0.003$, respectively; Table 5) but there was no significant interaction effect ($P = 0.46$; Table 5). The beak scores was higher in control and UVA/B hens than the UVA light treatment but increased after 20 weeks in all treatment groups. On the other hand, a significant interaction effect of light treatment and age was

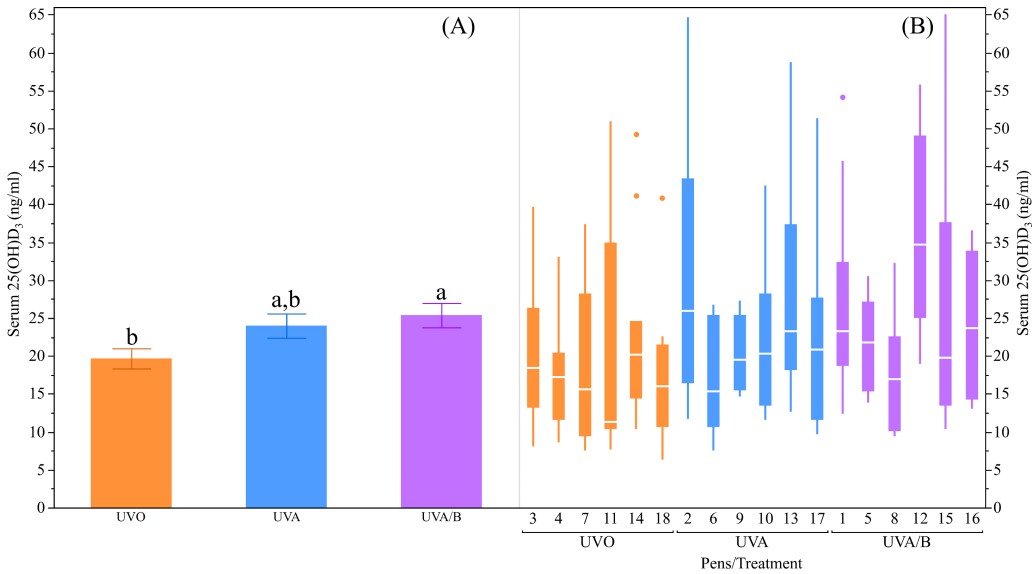

**Figure 3 The effects of light treatment (ultraviolet (UV) 0, UVA, UVA/B) on serum 25(OH)D$_3$ levels.** (A) The mean ± SEM effects between the treatment groups, (B) light effects across the pens within the same treatment group. The line within each box represents the median while the lower and upper boundaries of the box depict the interquartile range (*i.e.,* difference between the first and third quartiles). The whiskers extend to the outermost data point that falls within distances computed as follows: upper whisker = 3rd quartile + 1.5 × (interquartile range) and lower whisker = 1st quartile − 1.5 × (interquartile range). If data points do not reach the computed ranges, then the whiskers are determined by the upper and lower data point values (not including outliers). The disconnected points are outliers. Raw data are presented with analyses conducted on transformed data. [a,b]Dissimilar superscript letters indicate significant differences between the treatment groups ($P < 0.05$) as assessed by post-hoc Student's t-tests.

found on keel bone score ($\chi^2 = 14.33$, $df = 6$, $P = 0.03$; Fig. 4). Keel bone damage score increased across age irrespective of light treatment, but a higher score was documented in UVA/B hens at 20 and 24 weeks than in UVA hens. The keel bone scores were all similar among hens from different light treatments at 27 weeks (Fig. 4). There were no skin lesions or other issues observed except toenail damage in 5 and 11 hens under UVO and UVA/B treatments, respectively. All hens irrespective of light treatment had good feather coverage and no footpad damage at each observation point across the study period.

## DISCUSSION

This experiment aimed to investigate whether the supplementation of UVA/B light would have positive effects on improving egg production, egg and bone quality, and health during early lay in a controlled indoor setting relative to UVA only or no UV supplementation. The results showed that UVA/B light had minimal effects on egg production and egg quality, serum 25(OH)D$_3$, external health measures and bone quality. However, as expected, most parameters showed changes as the hens aged.

This study suggested there may have been some impacts of the UVA and UVA/B supplementation for increasing hen-day egg production (%) but that there was variation

Rana et al. (2023), *PeerJ*, DOI 10.7717/peerj.14997

**Table 5  Effects of UV light treatments on hens' body weight and external health scoring at 16, 20, 24, and 27 weeks.**

| Parameters | Treatment[1] | | | Age | | | | Test statistics (df, F-ratio, P-value) | | |
|---|---|---|---|---|---|---|---|---|---|---|
| | UVO | UVA | UVA/B | 16 weeks | 20 weeks | 24 weeks | 27 weeks | T | A | T x A |
| Body weight (kg) | 1.61 ± 0.02 | 1.63 ± 0.02 | 1.59 ± 0.02 | 1.35 ± 0.01[2] | 1.61 ± 0.01[c] | 1.70 ± 0.01[b] | 1.76 ± 0.01[a] | $F_{(2,14.28)} = 1.84, P = 0.19$ | $F_{(3,734)} = 2323.90, \boldsymbol{P < 0.0001}$ | $F_{(6,734)} = 0.68, P = 0.66$ |
| Toenail length (cm) | 1.40 ± 0.03 | 1.51 ± 0.03 | 1.41 ± 0.03 | 1.58 ± 0.04[a] | 1.35 ± 0.04[c] | 1.45 ± 0.04[b,c] | 1.38 ± 0.04[c] | $F_{(2,12.78)} = 3.00, P = 0.09$ | $F_{(3,746)} = 86.12, \boldsymbol{P < 0.0001}$ | $F_{(6,746)} = 1.26, P = 0.27$ |

| Parameters | Treatment[1] | | | Age | | | | Test statistics ($\chi^2$, df, P-value) | | |
|---|---|---|---|---|---|---|---|---|---|---|
| | UVO | UVA | UVA/B | 16 weeks | 20 weeks | 24 weeks | 27 weeks | T | A | T x A |
| Beak score[2] | 1.85 ± 0.02 | 1.78 ± 0.02 | 1.83 ± 0.02 | 1.84 ± 0.02 | 1.76 ± 0.03 | 1.80 ± 0.03 | 1.89 ± 0.02 | $\chi 2 = 4.83, df = 2, P = 0.09$ | $\chi 2 = 13.74, df = 3, \boldsymbol{P = 0.003}$ | $\chi 2 = 5.68, df = 6, P = 0.46$ |

**Notes.**

[1] Treatments: UVO, Control light with no ultraviolet (UV) spectral light; UVA, Control light plus UVA spectral light; UVA/B, Control light plus both UVA and UVB spectral light; T, Treatment; A, Age. The means ± SEM are presented for each variable. Raw data are presented with analyses conducted on transformed data. Significant *P* values (< 0.05) are indicated in bold.

[a–d] Dissimilar superscript letters indicate significant differences between the groups ($P < 0.05$).

[2] Score 0, 1, 2 indicates no, mild, moderate/severe damage, respectively.

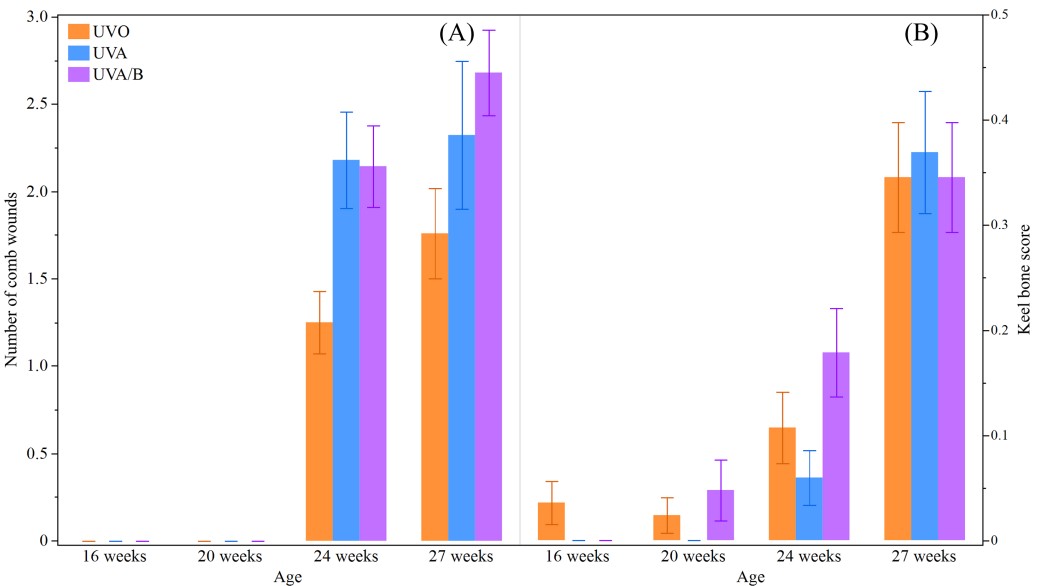

**Figure 4** **The mean ± SEM effects of light treatment (ultraviolet (UV) 0, UVA, UVA/B) on external health scores at different age points (16, 20, 24 and 27 weeks).** (A) Number of comb wounds, (B) Keel bone score (score 1, 2, or 3 indicates no, mild, moderate/severe damage, respectively). Raw data are presented with analyses conducted on transformed data (comb wounds).

across the different pens within the treatment groups. This variation could have resulted from different durations of time spent under the treatment lights across different pens, a factor to be quantified at the pen (not individual bird) level in the complimentary paper to this one (*Rana et al., 2021*, unpublished data). Previously, supplementation of UV light (either UVA and/or UVB) has not shown any significant effect on egg production during peak to mid-production phases (*Hogsette, Wilson & Semple-Rowland, 1997*; *Jones et al., 2001*; *Kühn et al., 2015*; *Sobotik, Nelson & Archer, 2020*; *Spindler et al., 2020*), with only a trend for increased egg production at 18 weeks of age in hens under natural forest light (containing UVA) over daylight (also containing UVA but differences in spectral intensity) (*Wichman et al., 2021*). Comparatively, UV light can sustain egg production during the later phase of the laying cycle (*Lewis, Ghebremariam & Gous, 2007*; *Wei et al., 2020*). This result, in conjunction with the finding that UV supplementation may accelerate sexual maturity by a couple of days warrants further exploration of whether UV light is beneficial during early lay, or if the most prominent benefits remain toward the end of the laying cycle.

The light treatments did significantly affect eggshell reflectivity (color) and yolk index where eggs from hens under UV lights had lighter eggshell color and a lower yolk index (indication of reduced egg freshness). Eggshell color depends on the amount of pigment (protoporphyrin IX) deposited in the eggshell during eggshell formation (*Samiullah, Roberts & Chousalkar, 2015*). These paler eggs under UV light are consistent with the reports of paler eggs in hens from free-range systems where the birds have access to sunlight containing UV radiation relative to indoor-housed birds (*Samiullah Roberts*

& *Chousalkar, 2014*; *Sekeroglu et al., 2009*). However, there are other factors including eggshell composition, nutrition and stress that can cause such color differences (*Liu & Cheng, 2010*). It is possible the UV treatments were causing stress responses in the hens, which could be supported by the increased comb wounds under the UV lights. Previous studies have found that UV supplementation reduced indicators of hen stress (*Sobotik, Nelson & Archer, 2020*), but did increase severe feather pecking (*Spindler et al., 2020*) while measures of heterophil/lymphocyte ratio in the same birds as the current study did not show treatment differences (*Rana et al., 2021*, unpublished data). The mechanisms underlying the deposition of protoporphyrin IX in relation to UV light warrant further investigation. Yolk index varies depending on egg weight and testing temperature (*Keener et al., 2006*; *Şekeroğlu & Altuntaş, 2009*). This study did not find any variation in egg weight across the light treatments and all eggs were very fresh so it is unclear why there may have been a higher index (greater freshness) in the control eggs or if the processing order on the day affected the yolk index parameter. None of the other tested egg quality traits were influenced by UV lights, similar to results reported in previous studies (*Kühn et al., 2015*; *Sobotik, Nelson & Archer, 2020*). The results suggest that exposure to UVA/B light, or at least intermittent exposure at certain intensities might not be sufficient to improve egg quality traits during early lay. Age did affect egg quality with parameters that either increased or decreased across time, consistent with age effects in previous experimental and commercial flocks (*Dikmen et al., 2017*; *Samiullah, Roberts & Chousalkar, 2015*; *Van Den Brand, Parmentier & Kemp, 2004*).

UVA/B light treatment did not show any significant effect on tibia bone health including tibia index, tibia breaking strength, ash content and mineral content. This may indicate that UVB exposure could not contribute additional value in bone growth and medullary bone formation during early lay. Medullary bone development coincides with pullet sexual maturity, and after onset of laying, medullary bone starts to contribute Ca for eggshell formation (*Fleming, 2008*; *Whitehead & Fleming, 2000*). For eggshell formation, around 60–70% of Ca comes from dietary feed with the rest from body stores (*Comar & Driggers, 1949*). During pre-lay and early lay (relative to end of lay) there might be an adequate source of calcium storage in the medullary bone and as a result, UVA/B light does not add additional value.

Similar to the results on bone parameters, the Ca and P plasma content were not affected by UVA/B light treatment, which is consistent with other studies (*Lietzow et al., 2012*; *Schutkowski et al., 2013*; *Wei et al., 2020*). However, similar to findings by *Lietzow et al. (2012)* there was a higher $25(OH)D_3$ concentration in hens under the UVA/B lights over the control hens but high variation between pens. The age-related inconsistencies between plasma Ca, P, and serum $25(OH)D_3$ may depend on the stage of egg formation in the hens' reproductive tracts and oviposition timing (*Nys & Le Roy, 2017*). While it was observed that the majority of hens had laid their eggs prior to sampling, it is unclear if circadian variation contributed to the inconsistences across age. Studies in broiler chickens (with or without Vit-$D_3$ dietary deficiencies) revealed that blood Ca improved when birds were exposed to UVB light (*Edwards Jr, 2003*; *Zhang et al., 2006*). This could be due to rapid absorption of Ca at a very early age, and/or feeding with Vit-$D_3$ supplementation in the diet

(*Edwards Jr, 2003*; *El-Safty et al., 2022*; *Lietzow et al., 2012*). Moreover, both plasma Ca and P content decreased across age, which was also likely a consequence of the higher demands on mineral metabolism towards peak egg production (*Habig et al., 2021*). Previous studies show the plasma concentration of $25(OH)D_3$ greatly increases when hens have access to dietary Vit-$D_3$ content (*Lietzow et al., 2012*; *Schutkowski et al., 2013*), however, birds in the current study were fed on a common commercial mash feed so any dietary Vit-$D_3$ effect on serum $25(OH)D_3$ was interpreted as consistent across treatment groups. The pen level variations suggest that the UVA/B light could stimulate Vit-$D_3$ synthesis, but is dependent on how long hens spend under the supplemental light. Therefore, the optimal timing and duration of exposure to ensure light supplement benefits needs to be further investigated.

There was no obvious difference in hens' body weight due to the UV light treatments, which supports other study findings (*Gongruttananun, 2011*; *Kühn et al., 2015*; *Schutkowski et al., 2013*; *Wichman et al., 2021*). There were more comb wounds in the UV hens than the control hens. The UVA spectrum is visually perceived by birds (*Lewis & Morris, 2000*; *Prescott & Wathes, 1999a*), and can be reflected from feathers, combs or other objects (*Prescott & Wathes, 1999b*; *Spindler et al., 2020*). Thus, the supplemental light appeared to increase conspecific comb pecking behavior, similar to previous reports of more skin injuries in laying hens under UVA supplementation relative to standard white fluorescent light (*Spindler et al., 2020*). However, there were no visible differences in plumage condition in the current study which is in contrast with increased feather pecking in the same previous commercial study (*Spindler et al., 2020*). It is possible that feather pecking behavior and, thus, plumage damage could have increased as the hens aged. Therefore caution does need to be applied for UVA supplementation if it has these negative consequences. Hens under UVA/B light had greater keel bone damage compared to hens under UVA lights at some earlier age points, but these differences disappeared by 27 weeks of age. It is unclear if this was a consequence of more perching in the UVA/B hens, or increased activity in the home pen—a query to resolve during the complimentary behavioral analyses (*Rana et al., 2021*, unpublished data). Further research in both experimental and commercial settings would be beneficial to verify behavioral pros and cons associated with more natural lighting conditions.

## CONCLUSION

During early lay, UV wavelengths, particularly UVA/B relative to UVA alone, had minimal effect on improving egg production, egg quality and bone health. This may be due to optimized nutrition from formulated feed and/or insufficient exposure when there is only one portion of the pen that is illuminated with UV radiation. UVA/B exposure may stimulate Vit-$D_3$ synthesis, but optimum duration and intensity needs to be determined to ensure this benefit. Future research identifying the relationship between UV exposure (UVA and UVB) and impacts on behavioral expression as well as long-term studies across the entire laying cycle would be of importance to quantify UV supplementation benefits in indoor laying hen production systems.

## ACKNOWLEDGEMENTS

The authors would like to thank the Animal House staff at the University of New England (UNE) for the experimental set-up and sampling assistance. The authors are also grateful to Jim M. Lea (CSIRO) and Tim R. Dyall (CSIRO) for their technical and husbandry assistance and to James Turnell, Craig Johnson and Brian Cross from UNE for their laboratory assistance and technical support. Thank you to Terence Sibanda (UNE) for his help in egg quality measures.

### Funding

The study was funded by the University of New England International Post-graduate Research Scholarship and the Commonwealth Scientific and Industrial Research Organisation (CSIRO) McIlrath Trust Scholarship to Md Sohel Rana, and Australian Eggs (Grant number: 31HS902CO). The funders had no role in study design, data collection and analysis, decision to publish, or preparation of the manuscript.

### Grant Disclosures

The following grant information was disclosed by the authors:
University of New England International Post-graduate Research Scholarship.
Commonwealth Scientific and Industrial Research Organisation (CSIRO) McIlrath Trust Scholarship.
Australian Eggs: 31HS902CO.

### Competing Interests

The authors declare there are no competing interests.

### Author Contributions

- Md Sohel Rana conceived and designed the experiments, performed the experiments, analyzed the data, prepared figures and/or tables, authored or reviewed drafts of the article, and approved the final draft.
- Jonathon Clay performed the experiments, authored or reviewed drafts of the article, and approved the final draft.
- Prafulla Regmi conceived and designed the experiments, authored or reviewed drafts of the article, and approved the final draft.
- Dana L.M. Campbell conceived and designed the experiments, performed the experiments, analyzed the data, authored or reviewed drafts of the article, and approved the final draft.

### Animal Ethics

The following information was supplied relating to ethical approvals (*i.e.*, approving body and any reference numbers):

The experimental design was approved by the Animal Ethics Committee of the University of New England (Approval no.: AEC21-034).

## Data Availability

Campbell, Dana; Rana, Sohel; Clay, Jonathon; Walkden-Brown, Stephen (2022): Egg production, egg and bone quality, and health of young laying hens under UV supplementation. v1. CSIRO. Data Collection. https://doi.org/10.25919/9660-rk44.

## Supplemental Information

Supplemental information for this article can be found online at http://dx.doi.org/10.7717/peerj.14997#supplemental-information.

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
