# Peer review of "Minimal effects of ultraviolet light supplementation on egg production, egg and bone quality, and health during early lay of laying hens"

_PeerJ, doi:10.7717/peerj.14997_

## Round 0.1 · original submission · Minor Revisions

The manuscript is interesting however for the broader interest of the readers, authors should include the data of UV on the hormonal profile of hens (FSH, LH, GnRH, GnIH).

Reviewer 1 ·

Basic reporting

This paper reports the findings of a study designed to assess the effects of UVA and UVB light supplementation on a variety of factors in laying hens, including egg quality and production, bone strength, hen growth and welfare. All sections of the paper are well written: appropriate background is given to provide context for the study; methods and results are cleared presented, and the conclusions are appropriate to the results of the study.

There are a couple of minor areas that could be improved:
Lines 90-92 - you introduce the terms 25(OH)D3, P, Ca, and 1,25(OH)2D3. It's not my area of expertise, but I didn't know what these were. Perhaps you could define and explain how they relate to vit D.
Figures 2b and 3b - points of the graph should not be joined by lines, since they are not connected datapoints, but values from different pens.
There are a couple of areas where clarity could be slightly improved in the intro. For example, in line 74 you conclude your paragraph by saying there may be effects during the early laying phase, but it is not obvious why the early laying phase is important from your summary of previous studies. Perhaps give a little more detail about those studies.
At line 79 could you put 'the osteoperosis process' in context? How many hens are affected and to what level of severity?
Throughout the introduction I wondered what the relevance would be for free range hens? For example, do hens use the range enough for the levels of light supplementation you mentioned to be relevant? Presumably you are thinking mostly about the positive benefits of supplementation for indoor hens. But if free range hens make limited use of the range, then their UVA/B exposure would not be comparable to treatment levels here? Doesn't affect your outcomes, but it would be good to understand the commercial context.

Experimental design

Research question is well defined, and it is clearly stated what knowledge gaps are addressed. Methods are clearly reported and the experimental design is robust and to an appropriate ethical standard.

As a minor point it may not be obvious to all what rearing houses with curtained sides are and how they are managed (line 91). Perhaps you could provide a description of these systems here.

Validity of the findings

Statistical analyses are clearly reported and robustly carried out. Conclusions are clear and sensible, and appropriate to the results.

Two points for consideration before publication:
1. Did you include experimental room as a random effect in your model? Some of the graphs suggest there may have been a room effect (e.g. lower production in the room which I assume contained pens 13-18). Could this explain some of the variability between pens? Would be good to explore in your analyses.
2. You don't mention the differences in keel damage between trt groups in your discussion. Why do you think these differences occurred?

Additional comments

Overall the study is really interesting and clearly presented. It provides useful information increasing the body of evidence in an under-studied field.

You mention several times in the discussion that pen level variation suggests differences in exposure to supplementary lighting, presumably because of differing perch use. Are you able to quantify this from behavioural observations to look for correlations at an individual hen level?

Reviewer 2 ·

Basic reporting

.

Experimental design

.

Validity of the findings

.

Additional comments

1. Add the novelty/add value of this work in the introduction section
2. Add more details for birds management in the M&M section
3. Add more recent references for 2021 and 2022, if any
4. The Ms meets the journal readership and merits the scientific requirements of the journal

---

## Round 0.2 · accepted · Accept

I am happy to recommend the manuscript for publication.

Reviewer 2 ·

Basic reporting

The Ms meet the journal readership and could be accepted, the authors revised the paper according to review made by the reviewers and in my opinion could be accepted

Experimental design

The Experiment design is right

Validity of the findings

The findings of this work is valid under similar experimental condition

Additional comments

No further comments